# Intensive Neurorehabilitation and Gait Improvement in Progressive Multiple Sclerosis: Clinical, Kinematic and Electromyographic Analysis

**DOI:** 10.3390/brainsci12020258

**Published:** 2022-02-12

**Authors:** Su-Chun Huang, Simone Guerrieri, Gloria Dalla Costa, Marco Pisa, Giulia Leccabue, Lorenzo Gregoris, Giancarlo Comi, Letizia Leocani

**Affiliations:** 1Experimental Neurophysiology Unit, Institute of Experimental Neurology-INSPE, Scientific Institute San Raffaele, 20132 Milan, Italy; huang.suchun@hsr.it (S.-C.H.); guerrieri.simone@hsr.it (S.G.); dallacosta.gloria@hsr.it (G.D.C.); pisa.marco@hsr.it (M.P.); 2Faculty of Medicine, University Vita-Salute San Raffaele, 20132 Milan, Italy; giulialeccabue95@gmail.com (G.L.); gregorislorenzo96@gmail.com (L.G.); comi.giancarlo@unisr.it (G.C.); 3Casa di Cura del Policlinico, 20144 Milan, Italy

**Keywords:** gait deficit, multiple sclerosis, neurorehabilitation, inertial sensor, surface electromyography

## Abstract

Background: Gait deficit is a hallmark of multiple sclerosis and the walking capacity can be improved with neurorehabilitation. Technological advances in biomechanics offer opportunities to assess the effects of rehabilitation objectively. Objective: Combining wireless surface electromyography and wearable inertial sensors to assess and monitor the gait pattern before and after an intensive multidisciplinary neurorehabilitation program (44 h/4weeks) to evaluate rehabilitation efficiency. Methods: Forty people with progressive multiple sclerosis were enrolled. Wireless wearable devices were used to evaluate the gait. Instrumental gait analysis, clinical assessment, and patient report outcome measures were acquired before and after the neurorehabilitation. Spatiotemporal gait parameters, the co-activation index of lower limb muscles, and clinical assessments were compared pre- and post-treatment. Results: Significant improvements after intensive neurorehabilitation were found in most of the clinical assessments, cadence, and velocity of the instrumental gait analysis, paralleled by amelioration of thigh co-activation on the less-affected side. Subjects with better balance performance and higher independence at baseline benefit more from the neurorehabilitation course. Conclusions: Significant improvements in gait performance were found in our cohort after an intensive neurorehabilitation course, for both quantitative and qualitative measures. Integrating kinematic and muscle activity measurements offers opportunities to objectively evaluate and interpret treatment effects.

## 1. Introduction

Multiple sclerosis (MS) is a complex autoimmune disease and is the first cause of disability in young adults [1]. Gait deficit is one of the clinical hallmarks of MS and greatly affects patient’s quality of life [2], particularly in people with the progressive phase of the disease [3]. Muscle weakness, spasticity, cerebellar disturbances, and sensory loss often merge together, resulting in complex patterns of gait impairment. Several standardized clinical scales, as well as timed walking tests and patient-reported outcome measures (PROMs) have been used to assess gait in people with MS [4,5,6]. For example, the Expanded Disability Status Scale (EDSS) in the range between 4 and 6.5 addresses the maximum walking distance of people with progressive MS (pwPMS); however, its low sensitivity to change limits its use to test treatment effects in the short term, including neurorehabilitation interventions [7]. On the other hand, timed measures, such as the 25 foot walk test, addressing speed as part of the MS Functional Composite, have been used in clinical trials, but do not address gait quality, which has important implications, such as energy cost, fatigue, and risk of falls [8,9,10,11].

Advances in sensor technology offer opportunities to assess and detect changes in gait quality [12,13]. The kinematics of gait, measured with accelerators, are sensitive biomarkers in pwPMS, as abnormalities in gait pattern occurs much earlier than the changes detected with EDSS [14]. Moreover, non-invasive surface electromyography (sEMG) allows to measure the pattern of muscle activation associated with spasticity—a frequent symptom in pwPMS [15,16]. Several features of spasticity, such as co-contraction and failure of voluntary activation, may emerge only in dynamic conditions such as gait and not at bedside assessment of passive movements. In that setting, it may be impossible to distinguish between increased muscle resistance to passive movements due to viscoelastic changes or to spastic reflex activity [17]. Simultaneous recording of sEMG and kinematics during ambulation can provide a complete picture of how the muscle activities and the integrations among muscle groups contribute to the final gait [15]. In this observational study, we used a combination of wireless sEMG and wearable inertial sensors to assess and monitor the gait pattern before and after an intensive multidisciplinary neurorehabilitation program of 4 weeks, in order to evaluate the value of objective gait measures as an additional outcome to assess the efficacy of neurorehabilitation.

## 2. Materials and Methods

### 2.1. Subjects

We examined gait performance in 40 consecutive pwPMS, attending an intensive neurorehabilitation program in our institute between September 2017 and June 2018. Patients were aged 18–65 years, with EDSS up to 6.5 (able to walk for at least 20 m safely with or without aids), without orthopedic conditions, depression, or cognitive involvement potentially interfering with compliance to testing. All data were collected as part of their clinical routine care, according to the Guideline of Good Clinical Practice [18]. All patients completed their rehabilitation program and provided written informed consent to use of their data for research purpose. All the data were anonymized prior to analysis.

The baseline assessments were performed one day before the beginning of the neurorehabilitation, while the post-intervention evaluations were performed one day after the end of the program to assure the subjects had sufficient time to rest.

### 2.2. Clinical Assessment

At hospital admission, pwPMS underwent a clinical examination, including EDSS; the Modified Ashworth Scale (MAS) for spasticity at the knee and ankle joints was determined [19]; and the Medical Research Council Scale (MRC) for strength of the anterior and posterior muscle groups of the thigh and leg [20], bilaterally, was ascertained. These scores were used to define the less-affected (LA) and more-affected (MA) side in each patient. Static balance performance was assessed with the Berg Balance Scale (BBS) [21]. Overall disability was quantified with the Functional Independence Measure (FIM) [22] and Modified Barthel Index (BIM) [23] and the risk of falls with the Conley Scale (CS) [24].

Mobility was measured with Timed Up and Go test (TUG) and 6-Minute Walk Test (6MWT), while the Timed 10-m Walking test (T10MW) was performed to assess the gait analysis (details described in Section 2.4). All the tests were conducted with shoes and walking aids (if needed) and identical equipment was used for both pre- and post-rehabilitation tests. The mobility exams were carried out in a flat 18-m hallway, which was a sufficient space to maintain a steady speed during the whole T10MW. For the TUG test, pwPMS were asked to stand up from a chair, walk for 3 m, make a 180° turn around a cone, and then walk back and sit on the same chair. The time needed to perform the test was recorded [25]. On the other hand, for the 6MWT, subjects were asked to walking back and forth in the hallway without rest and the total distance walked were recorded [26].

Furthermore, the following PROMs were collected: the Italian translation of the 12-Items MS Walking Scale (MSWS-12) [4,27]; MS Spasticity Scale-88 (MSSS-88) [28]; Fatigue Severity Scale (FSS) [29]; and Numeric Rating Scale of Spasticity (NRS) [30].

### 2.3. Patient-Centered Intensive Multidisciplinary Rehabilitation Program

The inpatient neurorehabilitation program was individualized by a multidisciplinary team (composed of neurologists, physiotherapists, psychologists, nurses, and speech therapists) based on the clinical evaluation at baseline, and an integrated rehabilitation protocol was designed taking into account muscular weakness, spasticity and balance deficiency, gait impairment, as well as cognitive functions. For each patient, the training protocol included aerobic training, resistance training, postural and balance training, training of activities of daily living (ADL), and cognitive rehabilitation. The items of each training were tailored designed by the multidisciplinary team. A physical therapist guided and accompanied the patients during the whole rehabilitation program. The intensive program consisted of multidisciplinary interventions of 18 h/week, including two 1-h sessions of physiotherapy per day from Monday to Friday, and one 1-h session on Saturday. The duration of the rehabilitation program was 4 weeks and the total active treatment time was approximately 44 h.

### 2.4. Gait Analysis

Gait analysis was assessed using simultaneous recording of the tri-axial acceleration (G-Walk, BTS bioengineering, Italy) and wireless sEMG (FREEEMG-1000, BTS bioengineering, Milan, Italy) while the pwPMS performed the T10MW. The active electrodes of sEMG were attached directly on the skin overlying the Rectus Femoris (RF), the long head of the Biceps Femoris (BF), Tibialis Anterior (TA), and Medial Gastrocnemius (GM), bilaterally [31]. Recorded signals were remotely transferred to a USB receiver. According to the standard T10MW procedure, patients were asked to complete the test at the maximum speed they could safely walk while the examiner manually measured the time of the test with a stopwatch. The walking test was performed twice, and the averaged results were used for further analysis.

The acceleration data were analyzed with G-Studio (G-Studio software, BTS bioengineering, Milan, Italy). Cadence, velocity, step length, duration of stance and swing phases (LA and MA separately) were quantified.

The sEMG data were processed with self-developed MATLAB scripts (MATLAB 2016b, MathWorks, Natickm, MA, USA). The raw sEMG data were first band-pass filtered between 10 to 500 Hz, full-wave rectified, then smoothed with a low-pass filter (3.5 Hz cut-off frequency) and normalized to the largest recorded value of each muscle after removing the resting value. Co-activation between agonistic–antagonistic muscle pairs (RF–BF and TA–GM) of the MA and LA sides were quantified as the co-activation index (CoI), which was defined as the overlapping area of the normalized EMG data divided by the duration of the overlapping [32]. To ensure the inter-subject comparison, CoI was calculated from the first five stable consecutive steps (the 2nd to the 6th step) of the T10MW.

### 2.5. Statistics

To test whether rehabilitation results showed a significant improvement in different gait parameters (spatiotemporal, Col, and clinical assessment), paired-sample *t*-tests were used. To explore whether the gait parameters are more sensitive in assessing neurorehabilitation results compared to EDSS, patients were separated into two groups based on the presence/absence of the EDSS reduction after treatment; independent and one-sample *t*-test (versus zero) were applied on the delta of the pre- and post-treatment parameters for inspecting the inter- and intra-group difference. To explore if the baseline clinical characteristics could influence rehabilitation outcome, correlations between the baseline clinical assessments and the pre- and post-rehabilitation differences were performed. Spearman’s R coefficient was used to assess the nonparametric parameters (EDSS, NRS, MAS, MRC, CS, MSWS-12, FSS, MSSS-88, BIM, FIM, BBS) while Pearson’s rho coefficient was used for continuous variables (TUG and 6MWT). To investigate the impact of the neurorehabilitation program on the relationship between muscle activities and kinematics, correlations with Pearson’s rho coefficient were performed among the pre- and post-rehabilitation difference in the CoI and spatiotemporal parameters. All the statistical analyses were performed with Prism 5 (GraphPad Software, Inc., San Diego, CA, USA) and the alpha-level was set at 0.05.

## 3. Results

### 3.1. Subjects

Demographic features of the pwPMS and pre-/post-treatment clinical assessments are shown in Table 1. Some of the baseline characteristics are part of a separate analysis already published [13]. After intensive neurorehabilitation, we found significant improvements in EDSS, FIM, BBS, MSSS-88, FSS, MSWS-12, TUG, 6MWT, T10MW, and, on the MA side, average MAS and sum MRC. No significant changes were found for BIM, CS, and NRS.

### 3.2. Pre- and Post-Treatment Comparisons of Gait Parameters

When assessing pre- and post-treatment performance considering the instrumented gait measures during T10MW, significant improvements were found for cadence (*p* = 0.0085) and velocity (*p* = 0.0004). No significant difference was found assessing the duration of the stance and swing phase, nor step length (complete results are listed in Table 2). Considering the muscle co-activation pattern, we found a significant reduction in CoI in the RF–BF pair on the LA side after rehabilitation (*p* = 0.0382), while no significant improvement was found for the GM–TA pairs (Table 3).

### 3.3. Improved Gait Parameters Also Found in Patients with Stable Post-Training EDSS

After intensive neurorehabilitation, 11 patients showed an EDSS reduction (i.e., decrease of more than 0.5 in EDSS after the neurorehabilitation program), 29 showed stable EDSS, with no patients showing EDSS increase. As we found significant improvements in several parameters when all the subjects were analyzed as a single group, we further investigated whether the aforementioned improvements were associated with EDSS reduction. Considering the spatiotemporal gait parameters assessed during T10MW, patients with decreased EDSS showed higher velocity rise compared to patients with stable EDSS, also in the presence of a significant improvement after treatment when considering time, velocity, and cadence. In addition, even for patients with stable EDSS, significant improvements were found in time and velocity as well, with a trend in cadence (Figure 1).

### 3.4. Correlations

First, the relationship between baseline characteristics and the outcome of instrumental gait measures was evaluated. Lower scores at the Conley Scale at baseline, indicating a lower risk of falls, were correlated with a higher improvement of gait velocity after treatment (r = −0.4404, *p* = 0.0045; Figure 2A). Higher baseline BIM values, indicating better overall independency, were correlated with CoI amelioration between the RF–BF pair on the LA side (r = −0.7271, *p* = 0.0172; Figure 2B). Furthermore, higher baseline BBS scores, reflecting a better static balance, were correlated with increased duration of the swing phase on the MA side (r = 0.3606, *p* = 0.0241; Figure 2C) as well as with increased step length (r = 0.4475, *p* = 0.0043) after rehabilitation (Figure 2D).

When assessing the relationships between post-training changes in kinematics and CoI, we found a CoI reduction in the GM–TA pair on the MA side to be correlated with prolonged duration of the swing phase in the contralateral side (r = −0.3266, *p* = 0.0485; Figure 3).

Last, we estimated the correlations between the post-rehabilitation changes in the instrumental gait parameters and the clinical assessments, including PROMs. The improvement in 6MWT was positively correlated with velocity improvement in the gait kinematics (r = 0.5641, *p* = 0.0002; Figure 4A). TUG improvement correlated with increased velocity (r = −0.3324, *p* = 0.0361; Figure 4B) and cadence (r = −0.4299, *p* = 0.0056; Figure 4C), as well as a decreased CoI of the RF–BF MA side (r = 0.6065, *p* = 0.0028; Figure 4D). Improvement in the time to perform the 10-m walking test correlated with that to perform TUG (r = 0.543, *p* = 0.003), but not with changes in the 6MWT or PROMs.

No correlation was found between the post-neurorehabilitation changes in PROMs and those of the instrumental gait parameters.

## 4. Discussion

In the present study, wireless wearable devices were used to quantitatively evaluate the effects of a 4-week intensive multidisciplinary neurorehabilitation program on instrumental measures of gait in a cohort of pwPMS with mild to moderate disability. Besides a significant improvement in the standard clinical assessments and PROMs, we also detected a significant amelioration of the kinematic gait parameters, in the presence of interesting interactions with the sEMG results.

We have previously demonstrated that the gait pattern of pwPMS is characterized by lower cadence, slower speed, smaller step length, and excessive muscle co-activation at both thigh and leg muscles in comparison to healthy controls [13]. The ambulation pattern in pwPMS is also characterized by a reduced swing phase and prolonged double support phase, which is similar to what has been reported in elderly subjects [33,34].

The present results showed improved kinematics paralleled with an amelioration of thigh co-activation on the less-affected side. One possible explanation may be that the improvement in gait performance depends on a better lower limb coordination and/or reduced spasticity. Another interpretation could be that the high co-activation in the lower limbs at baseline was, at least partly, the result and not the cause of the very low gait speed (0.8 m/s), which was significantly improved after neurorehabilitation. Furthermore, despite no significant change was found at the group level, a reduction in co-activation at the leg level was correlated with a prolongation of the contralateral swing phase in the more-affected side, implying that neurorehabilitation might help pwPMS to increase stability by reducing excessive muscle co-activation during ambulation.

The present findings also confirm the limitations of EDSS to measure treatment response, as previously suggested [35,36]. Our approach demonstrated that even in the absence of significant EDSS changes, a condition found in the majority of our patients, neurorehabilitation could still bring advantages to the gait performance. Besides objective parameters, we also found positive effects of neurorehabilitation on patient-reported outcomes addressing walking, spasticity, and fatigue. The improvements could be attributable to a combination of refinement of physical abilities and new strategies of resource management [37,38]. However, no direct significant correlation was found between PROMs and the instrumental gait parameters, suggesting that objective and subjective measures should be integrated [39].

The present results also bring some implications on the issue of patient selection for neurorehabilitation programs. Considering the baseline characteristics, we found subjects with a lower risk of falls (as from the Conley scale) to show better results in terms of cadence and velocity improvement of gait after neurorehabilitation. Subjects with better baseline static balance (as from BBS) showed greater improvements in terms of step length and time of the swing phase. Finally, subjects with higher levels of independency in everyday activities at baseline (estimated with BIM) displayed a higher reduction in thigh co-activation parameters on the LA side. To summarize, our data suggest that, among a cohort of pwPMS with overall high disability, subjects with better balance performance and higher independency may benefit more from such an intensive neurorehabilitation course. For patients with higher risk of fall, a lower improvement in gait measures may be explained with suboptimal duration of training addressing balance. It is possible that quantitative assessments, performed at baseline, can be useful to orientate patients into different kinds of rehabilitation programs [12].

One major limitation of the present study is that reproducibility of the instrumental gait analysis was not assessed at baseline. In the present study, the 50% cadence improvements found may be considered above the minimal detectable change (MDC) of the G-walk instrumented, timed 25-foot walk test [40] that was obtained in healthy subjects, thus not allowing direct comparisons. When considering a 20% improvement as the minimal important clinical difference (MICD), 38% of subjects showed a velocity improvement greater than MICD, while only 13% refinement in cadence reached the reported cutoff [41]. Decavel and colleagues used different equipment to measure gait in people with MS and reported a higher cutoff (34% improvement) [42]. With that cutoff, only a few patients would be considered as clinically improved in our study. From the above literature, the MDC can be very different when measured with different equipment (for example, MDC in cadence varies from 2.8 to 10.2 p/min), even when testing healthy subjects. We may expect a higher MDC in pwPMS due to greater intra- and inter-subject variability. Therefore, the reproducibility of the instrumental gait parameters measured with G-walk in pwPMS will be needed to determine if the observed improvements are clinically relevant.

Other limitations of the present study are a relatively small sample size, requiring a larger confirmatory pivotal study. Further, the present study did not perform follow-up assessments after the end of the program to evaluate the duration of the positive effects of the treatment. As it is not realistic to maintain such a level of physical training for outpatients, it is important to estimate if and how long the benefits remain. As the scope of the study was to evaluate the effect of the intensive multidisciplinary program, the study design also cannot disentangle the relationship between each sub-category of the neurorehabilitation and its final contribution to the overall gait.

## 5. Conclusions

To conclude, after an intensive multidisciplinary rehabilitation course, a significant improvement in gait performance was found in our PMS cohort, in both the quantitative and qualitative measures. Improved gait stability and balance could be the result of a reduced inappropriate muscular co-activation of the lower limbs, indicating the importance of integrating kinematic and muscle activity measurements to objectively evaluate and interpret treatment effects. As expected, subjects with better baseline performances are those who could benefit the most from neurorehabilitation, with possible implications for clinicians to optimize patient selection.

## Figures and Tables

**Figure 1 brainsci-12-00258-f001:**
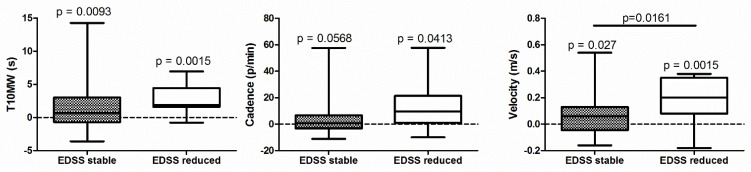
Changes in clinical (Timed 10-m Walking test—T10MW) and instrumented gait measures (cadence and velocity, measured with inertial sensors) in patients with stable (grey, N = 29) and reduced (white, N =11) EDSS after intensive neurorehabilitation. Data are presented as difference with respect to the baseline, with positive values indicating improvement. *p* values: one-sample *t*-test versus zero (above each box) or independent heteroscedastic *t*-test (above the line).

**Figure 2 brainsci-12-00258-f002:**
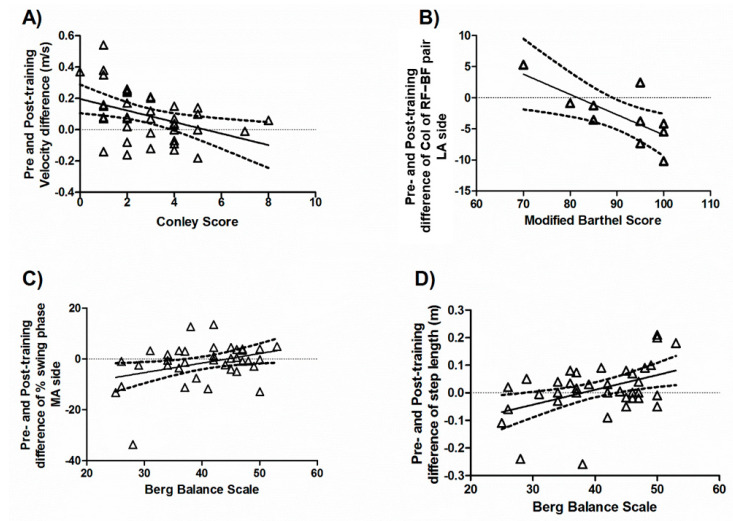
Correlation between the baseline clinical measures and the post-training improvements. Baseline scores of Conley Scale (**A**), Modified Barthel Index (**B**), and Berg Balance Scale (**C**,**D**) were correlated with changes of instrumental gait measures.

**Figure 3 brainsci-12-00258-f003:**
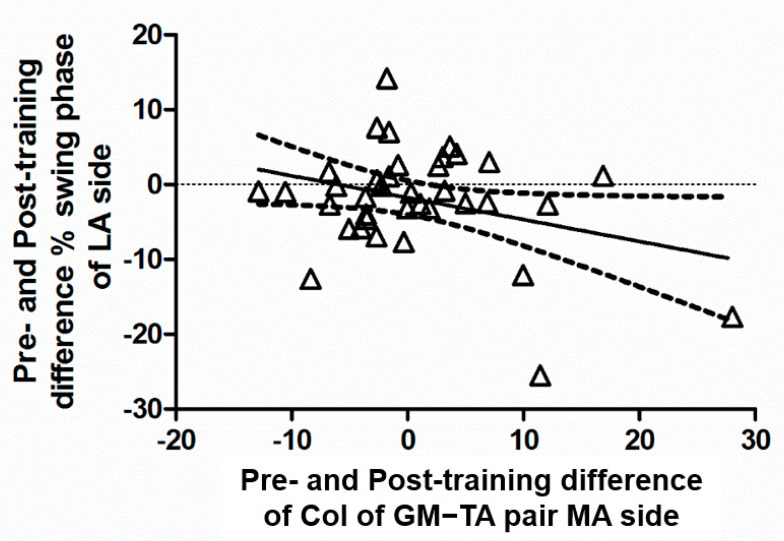
Correlation between changes in co-activation of the leg muscles in the more-affected side (MA, horizontal axis) and those in the swing duration of the less-affected side (LA, vertical axis). Positive values indicate a post-rehabilitation increase with respect to the baseline. GM–TA (Medial Gastrocnemius and Tibialis Anterior pair).

**Figure 4 brainsci-12-00258-f004:**
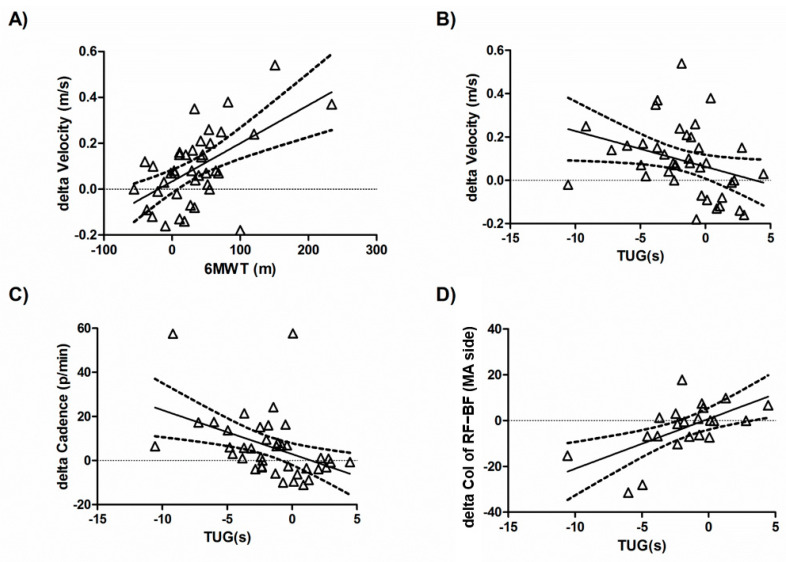
Correlation between changes in the clinical mobility assessments and those of the instrumental gait parameters from the sEMG and kinematic analyses. Positive delta values indicate post-rehabilitation increase with respect to the baseline. The improvement of 6MWT was positively correlated with velocity improvement (**A**). The TUG improvement correlated with increased velocity (**B**), cadence (**C**), as well as a decreased CoI of the RF–BF MA side (**D**). 6MWT (6-min Walk Test); TUG (Timed Up and Go Test); CoI (co-activation index); RF–BF (Rectus Femoris and long head of Biceps Femoris pair).

**Table 1 brainsci-12-00258-t001:** Patient demographics, clinical assessments, and PROMs at baseline and at the end of the intensive neurorehabilitation program.

Demographics	PMS Patients (n = 40)		
Gender (M/F)	20/20		
Age (y)	50.9 ± 9.8 (33–74)		
BMI	24.0 ± 4.6 (18.4–42.4)		
Disease Course	17% PP, 83% SP		
Disease Duration (y)	18.6 ± 10.1 (3.2–37.74)		
**Clinical Assessments**	**Baseline**	**Post-intervention**	** *p* ** **-value**
EDSS	6.0 (3.5–6.5)	6.0 (3.5–6.5)	0.0235 *
FIM	112.5 ± 9.00 (92–124)	114.6 ± 8.77 (92–126)	0.0006 *
BIM	88.4 ± 10.34 (65–100)	89.3 ± 9.37 (65–100)	0.1556
CS	2.9 ± 1.75 (0–8)	3.0 ± 1.58 (0–6)	0.3442
BBS	40.5 ± 7.68 (25–53)	45.6 ± 7.98 (23–56)	<0.0001 *
MRC (MA)	13.1 ± 3.25 (6–20)	14.2 ± 3.22 (8–20)	<0.0001 *
MAS (MA)	1.7 ± 1.38 (0–4)	1.5 ± 1.25 (0–4)	0.0086 *
TUG	16.6 ± 8.1 (7.4–35.5)	14.9 ± 7.0 (6.6–33.1)	0.001 *
6MWT	228.7 ± 95.6 (66–416)	262.8 ± 109.0 (77–504)	0.0001 *
T10MW	14.6 ± 7.35 (6.4–33.5)	12.7 ± 5.47 (5.7–26.6)	0.0003 *
**PROMs**	**Baseline**	**Post-intervention**	***p*-value**
NRS	3.9 ± 2.60 (0–8)	3.1 ± 2.54 (0–9)	0.0686
MSSS-88	188.6 ± 52.67 (104–278)	166.6 ± 58.2 (93–322)	0.0001 *
FSS	39.5 ± 15.00 (13.0–63.0)	34.8 ± 14.41 (12.0–65.0)	0.0079 *
MSWS-12	38.6 ± 9.73 (20.0–59.0)	35.0 ± 10.03 (19.0–58.0)	0.0106 *

Data are reported as the mean ± standard deviation (SD) and range (in bracket), except for EDSS (represented as the median and range). PMS (Progressive Multiple Sclerosis); BMI (Body Mass Index); PP (primary progressive); SP (secondary progressive); EDSS (Expanded Disability Status Scale); FIM (Functional Independence Measure); BIM (Modified Barthel Index); CS (Conley Scale); BBS (Berg Balance Scale); MRC (Medical Research Council Scale); MAS (Modified Ashworth Scale); TUG (Timed Up and Go Test); 6MWT (6-min Walk Test); T10MW (Timed 10-m Walking test); NRS (Numeric Rating Scale of Spasticity); MSSS-88 (MS Spasticity Scale-88); FSS (Fatigue Severity Scale); MSWS-12 (12-Item MS Walking Scale). An asterisk (*) denotes significant improvement after the rehabilitation.

**Table 2 brainsci-12-00258-t002:** Comparison of the spatiotemporal parameters pre- and post-training.

Kinematic Measures	Baseline	Post-Intervention	*p*-Value
Cadence (p/min)	98.9 ± 25.78 (41.8–158.3)	107.6 ± 21.00 (70.0–148.3)	0.0085 *
Velocity (m/s)	0.8 ± 0.33 (0.3–1.6)	0.9 ± 0.35 (0.4–1.8)	0.0004 *
% Time in swing phase (MA)	41.1 ± 5.00 (32.9–50.7)	40.7 ± 5.82 (25.9–51.3)	0.7605
% Time in stance phase (MA)	59.0 ± 5.00 (49.3–67.1)	59.3 ± 5.82 (48.7–74.1)	0.7586
% Time in swing phase (LA)	39.8 ± 5.58 (26.7–54.7)	38.8 ± 4.74 (29.2–47.8)	0.3069
% Time in stance phase (LA)	60.2 ± 5.55 (45.3–73.3)	61.2 ± 4.74 (52.2–70.8)	0.3188
Step length (m)	0.5 ± 0.11 (0.2–0.8)	0.5 ± 0.15 (0.2–0.8)	0.2995

Data reported as the mean ± SD (range). MA (more-affected side); LA (less affected side). An asterisk (*) denotes significant improvement after the rehabilitation.

**Table 3 brainsci-12-00258-t003:** Pre- and post-training co-activation index.

CoI	Baseline	Post-Intervention	*p*-Value
GM–TA pair (MA)	17.0 ± 6.87 (9.0–48.3)	15.8 ± 5.45 (7.2–30.1)	0.1956
GM–TA pair (LA)	19.3 ± 8.03 (7.4–47.3)	19.3 ± 8.13 (8.0–47.25)	0.4874
RF–BF pair (MA)	22.7 ± 11.56 (11.0–47.5)	19.4 ± 7.16 (8.3–35.1)	0.1020
RF–BF pair (LA)	20.9 ± 5.05 (15.2–31.6)	18.1 ± 4.07 (11.9–24.3)	0.0382 *

Data presented as the mean ± SD (range). CoI (co-activation index); GM–TA (Medial Gastrocnemius and Tibialis Anterior pair); RF–BF (Rectus Femoris and long head of Biceps Femoris pair); MA (more-affected side); LA (less-affected side). An asterisk (*) denotes significant improvement after the rehabilitation.

## Data Availability

The data are not publicly available due to patient confidentiality. The data presented in this study are available upon reasonable request from the corresponding author.

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
