# Peer review of "Intensive Neurorehabilitation and Gait Improvement in Progressive Multiple Sclerosis: Clinical, Kinematic and Electromyographic Analysis"

_brainsci, 2022, doi:10.3390/brainsci12020258_

Round 1
Reviewer 1 Report
Comments
General comments: the study is interresting by showing new approach in the assessment in MS to evaluate improvement after rehabilitation programm.
However, I have some comment to increase the impact of the study.
1°Introduction: the aim of the study is not clearly exposed
2°It is an observational study, please specify it.
Materials and methods:
3°Was the necessary number of patients calculated? If yes, specify the method.
4°Could the authors explain when the assessments were performed: before the beginning of the rehabilitation program (the first day of the program? And the second assessment? Dis the patients had sufficient time of rest before performing the second assessment?)
5°Line 70: is the risk of fall an exclusion criterion? If yes, specified it, if not, this evaluation has to be in the assessment part.
6°Line 77 and 78: how the clinicians could be able to isolate biceps femoris spasticity (and exclude ½ tendinous and ½ membranous one). In the same line, how the clinicians could be able to isolate the medial gastrocnemius spasticity from the triceps surae (I can imagine that the Ashworth scale was performed with extended knee). I have the same require for the strength of medial gastrocnemius muscle.
7°Line 83: please give some references for TUG and 6 MWT or explain your method to perform those tests.
8°Line 104: it is important to describe the method of the 10MW assessment. Did patient used shoes or barefoot? Please describe the area used: was it sufficient space before and after the 10 meters to be sure to measure the stabilized speed and not acceleration and deceleration.
9°Line 105: please give a reference for the placement of the electrodes (for exemple https://doi.org/10.1016/j.jelekin.2019.102363)
Results:
10°Had all the patients included finished the program? If yes, specified, if no describe the excluded patients. This is important in terms of faisability.
Discussion:
11°Some results showed improvement statistically significant. But what about the reproducibility. Where the improvement inside or the variability of spatiotemporal parameters? Since there was inly one pre-intervention assessment, the authors could refer to literature (Gait tests in multiple sclerosis: Reliability and cut-off values by Decavel et al.)
12°Line 232: be careful in the interpretation of increase in co-activation, this can be due to the increase of speed. 0.8m/s is very slow and induce an increase in co-activation by the decrease of velocity in articulations. Please modulate the interpretation.
13°Patients with lower risk of fall shows better results by the program. Could the authors propose another program more specifically based on balance to improve patients with higher risk of fall with different goals. Also assessments will be useful to orientate patient in different kind of rehabilitation program.
Author Response
First, we would like to thank you for spending your valuable time reviewing our paper. Thank you very much for the useful and insightful comments. Please find our point-by-point responses below:
- We re-phrased the last sentence of the introduction, hope it became more clear now, thank you.
- The description was added to the last sentence of the introduction.
- No, the number needed in the present study was based on an empirical assumption. Therefore, we did not calculate the sample size.
- The information was added to the last part of Materials and Methods-2.1 subject, thank you.
- Sorry for the confusion. It has been moved to the assessment part (Materials and Methods: 2.2 clinical assessment).
- Sorry for the incorrect description. It has been fixed.
- The description and references have been added to the second paragraph of 2.2 clinical assessment (Materials and Methods), thank you.
- The information has been added to the second paragraph of 2.2 clinical assessment, thank you.
- The reference has been added, thank you.
- The information has been added to the first paragraph of Material and Method (2.1 subjects), thank you.
- Thank you for pointing out this important issue. We reviewed related literature and added a paragraph in Discussion (paragraph 6) to address it as a major limitation of the current study. We will also keep in mind and add reproducibility assessment into our future studies to better investigate this issue, thank you very much.
- Thank you for providing an alternative explanation, the interpretation has been modified accordingly (Discussion, paragraph 3).
- Thank you for the important comments, also the other reviewers addressed a comment relating to this issue. The lack of significant improvements in patients with worse baseline performance may be due to the duration of the training was not long enough, or the program was not specifically designed to improve the balance. We combined your suggestions and added them in paragraph 5 of Discussion, thank you.
Reviewer 2 Report
This manuscript is well-written and results are presented clearly for the reader. There are only a few minor comments/suggestions from this reviewer.
Line 18-In the abstract it would be beneficial to provide context after the word “intensive” as this word is open to wide interpretation among rehabilitation professionals. I suggest putting (6 days a week/4 weeks) or (42hours/4weeks) after the word “intensive” or include a separate sentence describing the amount of neurorehabilitation received over the time period of 4 weeks, as this important element is currently missing from the abstract.
Line159 remove the word “instead” from the sentence as it is unnecessary.
Section 3.3. While it is completely appropriate to subdivide patients into those that showed a decrease of more than 0.5 in EDSS (11 patients) and those that showed no change in EDSS) and present significant findings from comparisons between these subdivided groups. It did not appear that this subdivision was predetermined prior to onset of data collection, rather the original intent was to evaluate pre- vs post-intensive rehab as a single group (without dividing into “responders” vs “nonresponders”) It would beneficial to address pre-vs post intensive rehab findings as a single group (and if there were no significant differences, proceed to divide the group based on improvement in EDSS vs stable EDSS (no change in EDSS) scores similar to how Line 237 addressed this aspect regarding no change at the group level.
Line 211 “10 meter walk test correlated “with” that to perform
Line 218-219. It would be beneficial to briefly address this finding further in the discussion section.
Lines 260-266 It should be recognized as a limitation that there was no follow-up testing after the end of the 4 weeks of intensive neurorehabilitation. It is a very real possibility that the significant improvements (even among the subjects with better balance and higher independency at baseline) demonstrated would be retained only for a short time after the intensive 6 days/week -2hrs a day neurorehabilitation ended. Future studies should include a follow-up evaluation of study participants as this level of intense rehabilitation is likely not realistic to be sustained over prolonged periods of time (months).
In the discussion, while it is acknowledged that subjects with better balance and higher independency at baseline benefited more from intensive neurorehabilitation, one would think individuals with poorer baseline assessments might be the individuals in greatest need of intensive neurorehabilitation, but perhaps these individuals just require a longer period of intensive rehabilitation than just 4 weeks to see significant benefit. This could be briefly addressed in the discussion of limitations and future studies.
Author Response
Thank you very much for spending your valuable time reviewing our paper, and thank you for the compliments. We really appreciate your comments and have modified the manuscript accordingly. Please find the point-by-point response below:
- We fully agree and are sorry for did not include it in the abstract. Also, the clinician pointed out that the duration of the session on Saturday for physical training was 1 hour rather than thirty minutes. Therefore, the resulted total physical training time was 44 hours/week. The information has been fixed in the text and added to the abstract (line 5), thank you.
- Thank you very much, it has been removed.
- The single-group analysis was performed in 3.1 and 3.2 and we found improvement at the group level. As we also found a sub-set of patients who showed a reduction of EDSS, we decided to further explore if the improvements were found only in those with EDSS reduction. We were surprised when we found that the refinements also existed in patients with steady EDSS. Since this analysis was not decided prior to the study design, it was not described in the introduction. We understood that this may cause confusion during reading and added 1 sentence in section 3.3 to address this issue. Hope that this way clarifies the confusion, thank you.
- Thank you very much and we are sorry for not being more careful during the English editing. It has been corrected.
- One sentence has been added to the end of paragraph 4 in Discussion to address this finding.
- Thank you very much for pointing this issue out. It is indeed a very important issue and we agree that this is a limitation of the current study. We added this issue in the last paragraph of Discussion and will improve the design of future studies to be able to evaluate it.
- Thank you very much for this important comment, also the other reviewer addressed a comment relating to this issue. The lack of significant improvements in patients with worse baseline performance may be due to the duration of the training was not long enough, or the program was not specifically designed to improve balance. We combined your suggestions and added them in paragraph 5 in Discussion, thank you.
Round 2
Reviewer 1 Report
The document is now much clearer and the limitations well identified.